# *NeuralFluid* : Nueral Fluidic System Design and Control with Differentiable Simulation

**Yifei Li**
MIT CSAIL

**Yuchen Sun**
Georgia Institute of Technology

**Pingchuan Ma**
MIT CSAIL

**Eftychios Sifakis**
University of Wisconsin-Madison

**Tao Du**
Tsinghua University, Shanghai Qi Zhi Institute

**Bo Zhu**
Georgia Institute of Technology

**Wojciech Matusik**
MIT CSAIL

## Abstract

We present *NeuralFluid* , a novel framework to explore neural control and design of complex fluidic systems with dynamic solid boundaries. Our system features a fast differentiable Navier-Stokes solver with solid-fluid interface handling, a low-dimensional differentiable parametric geometry representation, a control-shape co-design algorithm, and gym-like simulation environments to facilitate various fluidic control design applications. Additionally, we present a benchmark of design, control, and learning tasks on high-fidelity, high-resolution dynamic fluid environments that pose challenges for existing differentiable fluid simulators. These tasks include designing the control of artificial hearts, identifying robotic end-effector shapes, and controlling a fluid gate. By seamlessly incorporating our differentiable fluid simulator into a learning framework, we demonstrate successful design, control, and learning results that surpass gradient-free solutions in these benchmark tasks.

## 1 Introduction

Complex fluidic systems play an important role in many engineering and scientific disciplines, encompassing applications at different length scales ranging from biomedical implants [1], microfluidic devices [2], hydraulic devices to and flying robots [3]. Understanding these fluid-solid coupling mechanisms in nature and mimicking their control strategies in artificial designs is essential for advancing our control and design capabilities to synthesize novel solid-fluid systems.

Devising neural control algorithms to accurately manipulate the behavior of a complex fluidic system and optimize its performance remains challenging due to the intricate interplay between device geometry, control policies, flow dynamics, and the inherent physical and optimization constraints unique to each fluidic system. On one hand, differentiable simulation fluid-system interactions are inherently difficult because simulation is dynamic, involving a sequence of forward and backward steps interleaved with control signals that are computationally expensive. On the other hand, naively employing traditional control algorithms, mainly derived from their solid counterparts, to control fluidic systems remains difficult due to characterizing the infinite degrees of freedom of fluid flows and their interactions with solid boundaries. The co-design of fluid-solid systems, involving both shape and control, is critical to exploring the optimal performance of these systems.

Currently, the machine learning community lacks a computational Gym-like [4] environment to facilitate the exploration of fluidic systems manifesting strong solid-fluid interactions and controllable

38th Conference on Neural Information Processing Systems (NeurIPS 2024).

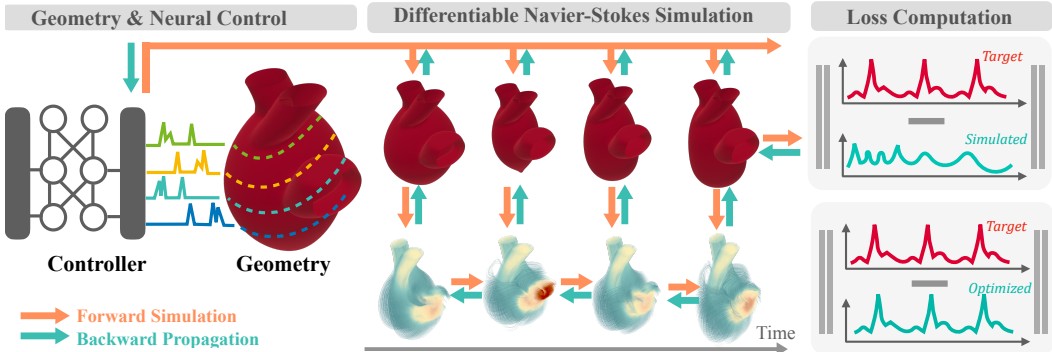

Figure 1: **Pipeline Overview.** (1) Our pipeline starts with an initial parametric geometry and a neural network parameterized controller. (2) The fluid dynamics is then simulated using a dynamic Navier-Stokes solver. (3) The performance of the design and control is evaluated using a loss function, the gradients of which are then back-propagated through our end-to-end differentiable framework. (4) The gradient-based optimization iteratively improves the geometry and control to achieve the task goal. This pipeline allows for efficient geometry and control co-optimization.

dynamic boundaries. Recent literature in robotic learning (e.g., [5]) has established unified multi-physics differentiable simulation platforms to facilitate learning control policies for various fluid interactions in daily scenarios. Similar ideas can be observed in [6, 7, 8], where differentiable simulation plays a central role in accommodating various design and optimization tasks of dynamic systems involving fluid dynamics. However, despite these inspiring advances, learning the control policies and exploring the optimal performance of a dynamic fluidic system with complex boundary conditions remains difficult due to their inherent complexities in differentiating solid boundary behaviors and optimizing their fluidic consequences due to these boundary motions.

This paper presents a novel framework for a fully automated pipeline aimed at devising neural controls for complex fluidic systems with dynamic boundaries. Our framework is designed to robustly control complex fluidic systems that consist of externally driven soft boundaries and internal complex flow behaviors, such as those systems underpinning an artificial heart or a microfluidic device.

*NeuralFluid* consists of three critical components to enable neural control of a complex fluidic system. First, we devise a differentiable geometry representation to offer an expressive design space while remaining low-dimensional, enabling efficient exploration by the optimization algorithm. Second, we implement a differentiable fluid simulator with solid-fluid interface handling to accurately characterize the dynamic fluid behavior and predict its spatiotemporal impact on the moving boundaries. We back-propagate gradients at the solid-fluid interface to extend gradient computation to the geometry iso-surface. Last, we provided an optimization framework to efficiently search the design space, considering the underlying fluid dynamics and boundary conditions.

Our pipeline features a low-dimensional parametric geometry representation capable of expressing complex shapes and a differentiable Navier-Stokes simulator with geometry gradient computation for predicting dynamic fluid behavior in response to control signals. In addition, our pipeline leverages gradient-based optimization for efficient design space exploration, co-optimization of the device geometry and control, and accurate performance evaluation of the design under dynamic flows. To showcase the practical implications and versatility of our approach, we have established a suite of Gym-like [4] environments. These benchmarks are designed to test applications in robotics and engineering, facilitating advancements in system identification, optimization of end-effector shapes and controls, and the dynamic optimization of structures such as artificial hearts within a closed-loop control framework. We showcase the effectiveness of our pipeline in facilitating different design and control tasks, including amplifier, fluidic switch, flow modulator, shape and position identification, closed-loop control of water gate and artificial heart.

We summarize our main contributions as follows:

- Development of a fast differentiable Navior-Stokes simulator for optimization in 2D and 3D scenes.

- Development of a low-dimensional differentiable parametric geometry representation for complex shapes embedded into the differentiable simulation pipeline.

- Gradient computation extension to geometry iso-surface to enable control and geometry co-design and iso-surface optimization.

- Gym-like [4] environments and benchmarks to demonstrate applications in robotics and engineering, including the design of amplifier, fluidic switch, flow modulator, geometry system identification, and closed-loop control of a fluid gate and artificial heart.

## 2 Method

### 2.1 Pipeline Overview

We present an overview of our method in Fig. 1. Our pipeline defines the designs with a low-dimensional parametric geometry representation (Sec. 2.2). The behavior and performance of the design in the fluid environment are evaluated by a dynamic differentiable Navier-Stokes simulator (Sec. 2.3). Both components are embedded in a gradient-based optimization framework that co-optimizes both the geometric design and the control signal until convergence.

### 2.2 Geometry Representation

We represent our geometry with a low-dimension representation. Take the illustration in the inset figure as an example, here we introduce the representation on a high level, and refer the readers to the appendix for the full details. We parameterize a closed 2D surface using its center $\mathbf{c}$ and a set of connected Bezier curves with their control points de-

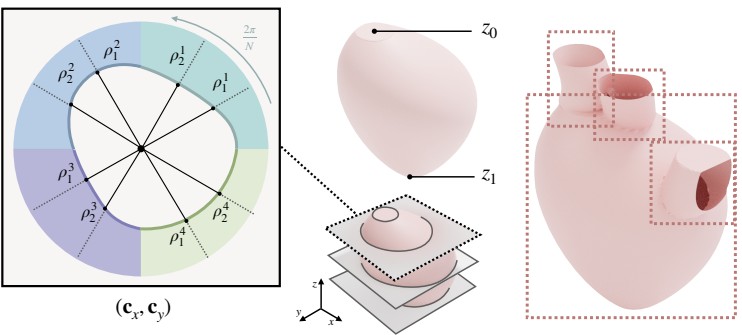

a) Closed 2D Surface      b) Closed 3D Surface      c) 3D Geometry

fine in polar coordinates $\rho_i$ for $i \in [1, 2, \ldots, 2N]$, where every two control points define a 2D Bezier curve spanning $\frac{2\pi}{N}$ radians in the polar coordinate system. This representation offers a compact way of defining diverse geometries. We further parameterize a closed 3D surface using a list of 2D surfaces defining the key cross-sections of the geometry along an extrusion axis $z$ of the local object frame, where each 2D surface is parameterized as described above. The parameterization includes $z = z_0$ and $z_1$, which determines the Z plane of the first and last cross-section, along with the parameters for each key 2D cross-section, which are assumed to be evenly spaced between $z \in [z_0, z_1]$. The continuous geometry interpolates the key cross-sections along the z-axis. Finally, we construct more complex 3D geometries using operations from Constructive Solid Geometry (CSG): Union and intersection, which allows us to define a 3D parametric heart model using the union of four sub-geometries.

### 2.3 Differentiable Navier-Stokes Simulation

Our fluid dynamics is governed by the incompressible Navier-Stokes equations. These consist of the momentum equation (Eq. 1a), accounting for temporal changes in velocity ($\boldsymbol{u}$), advective acceleration, viscous dissipation, and pressure ($p$) gradient forces for an incompressible fluid with fluid density $\rho$ and kinematic viscosity $\nu$. The incompressibility condition (Eq. 1b) requires the divergence of the velocity field must be zero to enforce the conservation of mass:

$$
\begin{cases}
\dfrac{\partial \boldsymbol{u}}{\partial t} = -(\boldsymbol{u} \cdot \nabla)\boldsymbol{u} + \nu \nabla^2 \boldsymbol{u} - \dfrac{1}{\rho}\nabla p, & \text{(1a)} \\
\nabla \cdot \boldsymbol{u} = 0 & \text{(1b)}
\end{cases}
$$

### 2.3.1 Numerical Simulation

We build the fluid simulator by leveraging the operator-splitting method [9][10]. A single simulation step comprises three sub-steps: advection, viscosity, and projection. See the Appendix B for details on time discretization. The simulation domain is discretized on a standard Marker-and-Cell (MAC) grid [11], with pressures stored at cell centers and velocities at cell faces. By employing the finite-difference scheme on the MAC grid cells and faces, we construct the matrix $\frac{1}{\Delta x}\mathbf{G}$ for gradient operator and its negative transpose $-\frac{1}{\Delta x}\mathbf{G}^T$ for divergence operator. In the following sections, capital letters will refer to matrices or the flattened vectors induced by the fields denoted by the corresponding lowercase letters in Appendix B.

**Advection**  We employ the semi-Lagrangian advection scheme, where the advected velocity field $\tilde{\mathbf{U}}^{n+1}$ is a linear interpolation of the velocity field $\mathbf{U}^n$. The interpolation position function is a function of $\mathbf{U}^n$ can be put into a matrix form $\mathbf{B}$, which results in:

$$\tilde{\mathbf{U}}^{n+1} = \mathbf{B}(\mathbf{U}^n)\mathbf{U}^n. \tag{2}$$

**Viscosity**  For incompressible fluid with a constant viscosity coefficient, the viscous force density is the product of the Laplacian of velocity and the viscosity coefficient. For each axis, the Laplacian of the corresponding velocity component is calculated on grid points using the finite difference method:

$$\hat{\mathbf{U}}^{n+1} = \left(\mathbf{I} - \frac{\nu\Delta t}{\Delta x^2}\mathbf{G}^T\mathbf{G}\right)\tilde{\mathbf{U}}^{n+1}. \tag{3}$$

**Projection**  The projection step ensures the incompressibility of the fluid. Solid un-aligned with the grid may intersect with grid faces, which can be captured with a cut-cell method [12]. We introduce $\alpha^{n+1}$ to represent the fluid proportion of a grid face. The solid's signed distance function (SDF) $\phi^{n+1}$ and the velocity $\boldsymbol{u}_s^{n+1}$ can derived from the solid geometry. We first use marching cube to compute the geometry zero contour. Next, we identify the intersection points between this contour and the grid faces and compute $\alpha^{n+1}$ based on $\phi$. For instance, in the inset figure, grid face $(i+\frac{1}{2}, j, k)$ is cut by the contour, then $\alpha_{i+\frac{1}{2}, j, k}^{n+1} = \frac{S_{AEF}}{S_{ABCD}} = \frac{1}{2} \cdot \frac{|AE|}{|AB|} \cdot \frac{|AF|}{|AD|} = \frac{1}{2} \cdot \frac{\phi_A^{n+1}}{\phi_A^{n+1} - \phi_B^{n+1}} \cdot \frac{\phi_A^{n+1}}{\phi_A^{n+1} - \phi_D^{n+1}}$.

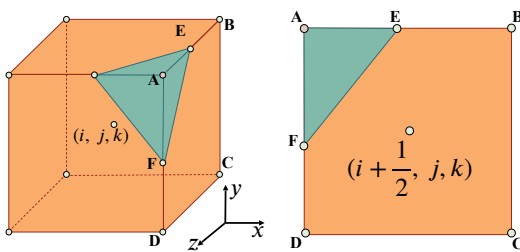

The volume change rates for fluid and solid at grid cell $(i, j, k)$, denoted as $\gamma_{f,i,j,k}^{n+1}$ and $\gamma_{s,i,j,k}^{n+1}$ respectively, equal to the sum of flux on the cell's surrounding faces, which can be calculated using $\alpha^{n+1}$, fluid velocity $\boldsymbol{u}^{n+1}$, and solid velocity $\boldsymbol{u}_s^{n+1}$.

The incompressibility condition gives requires the sum of $\gamma_{f,i,j,k}^{n+1}$ and $\gamma_{s,i,j,k}^{n+1}$ to be zero, which gives

$$\frac{\Delta t}{\rho\Delta x}\mathbf{G}^T\mathbf{S}^{n+1}\mathbf{G}\mathbf{P}^{n+1} = \mathbf{G}^T\mathbf{S}^{n+1}\hat{\mathbf{U}}^{n+1} + \mathbf{G}^T(\mathbf{I} - \mathbf{S}^{n+1})\mathbf{U}_s^{n+1}. \tag{4}$$

where $\mathbf{S}^{n+1}$ is a diagonal matrix induced by $\alpha^{n+1}$, and $\mathbf{P}$ is the pressure. After solving the linear system, the fluid velocity is updated based on the pressure values:

$$\mathbf{U}^{n+1} = \hat{\mathbf{U}}^{n+1} - \frac{\Delta t}{\rho\Delta x}\mathbf{G}\mathbf{P}^{n+1}, \tag{5}$$

### 2.3.2 Back-propagation through Time

We construct our back-propagation algorithm to mirror the sequence of operations carried out in the forward pass but in a reversed order.

Given the gradients of the loss function $J$ with respect to the velocity field $\boldsymbol{u}$ at time step $n+1$, denoted by $\frac{\partial J}{\partial \mathbf{U}^{n+1}}$, our goal is to compute the corresponding gradients at time step $n$, $\frac{\partial J}{\partial \mathbf{U}^n}$.

**Projection**  We begin by reversing the projection step to back-propagate $\frac{\partial J}{\partial \mathbf{U}^{n+1}}$ to derive $\frac{\partial J}{\partial \mathbf{P}^{n+1}}$ and $\frac{\partial J}{\partial \hat{\mathbf{U}}^{n+1}}$. Back-propagating through Eq.5 gives

$$\frac{\partial J}{\partial \mathbf{P}^{n+1}} = -\frac{\Delta t}{\rho \Delta x} \frac{\partial J}{\partial \mathbf{U}^{n+1}} \mathbf{G}. \tag{6}$$

We can back-propagate the adjoint of Eq. 4 w.r.t $\hat{\mathbf{U}}^{n+1}$ by defining the adjoint variable $\mathbf{y}$ and derive

$$\frac{\partial J}{\partial \hat{\mathbf{U}}^{n+1}} = \frac{\partial J}{\partial \mathbf{U}^{n+1}} + \mathbf{y}\mathbf{G}^T\mathbf{S}^{n+1}, \tag{7}$$

where $\mathbf{A} = \frac{\Delta t}{\rho \Delta x}\mathbf{G}^T\mathbf{S}^{n+1}\mathbf{G}$, $\mathbf{b} = \mathbf{G}^T\mathbf{S}^{n+1}\hat{\mathbf{U}}^{n+1} + \mathbf{G}^T(\mathbf{I}-\mathbf{S}^{n+1})\mathbf{U}_s^{n+1}$, and $\mathbf{y}$ is computed by solving the linear system $\mathbf{A}y^T = (\frac{\partial J}{\partial \mathbf{P}^{n+1}})^T$.

**Viscosity and Advection**  Back-propagating through viscosity and advection simply involves back-propagating $\frac{\partial J}{\partial \hat{\mathbf{U}}^{n+1}}$ through Eq. 3 and Eq. 2, which allows us to derive:

$$\frac{\partial J}{\partial \mathbf{U}^n} = \frac{\partial J}{\partial \hat{\mathbf{U}}^{n+1}} \left( \mathbf{I} - \frac{\nu \Delta t}{\Delta x^2}\mathbf{G}^T\mathbf{G} \right) . (\frac{\partial \mathbf{B}}{\partial \mathbf{U}^n}\mathbf{U}^n + \mathbf{B}(\mathbf{U}^n)). \tag{8}$$

The above equations provide the outline of the back-propagation process through a single time step of from time step $n+1$ to $n$. To compute the gradients of the loss function $J$ at any time step, we iterate the back-propagation process over the full sequence of time steps.

### 2.3.3  Back-propagation through Geometry

The parametric geometry affects simulation through the solid-fluid boundary during the projection step in Eq. 4. Specifically, the SDF of the geometry $\phi^{n+1}$ affects the volume matrix $\mathbf{S}^{n+1}$ and the velocity (in the case of moving geometry) of the geometry $\mathbf{U}_s^{n+1}$ affects the boundary condition. We can back-propagate $\frac{\partial J}{\partial \mathbf{P}^{n+1}}$ w.r.t these two parameters to derive

$$\begin{cases} \dfrac{\partial J}{\partial \mathbf{S}^{n+1}} = y\mathbf{G}^T(\hat{\mathbf{U}}^{n+1} - \mathbf{U}_s^{n+1}) + y\dfrac{\partial \mathbf{A}}{\partial \mathbf{S}^{n+1}}\mathbf{P}^{n+1}, & \text{(9a)} \\[3mm] \dfrac{\partial J}{\partial \mathbf{U}_s^{n+1}} = -y\mathbf{G}^T\mathbf{S}^{n+1}. & \text{(9b)} \end{cases}$$

Further back-propagating $\frac{\partial J}{\partial \mathbf{S}^{n+1}}$ first through the SDF $\phi$ then through the distance computation and $\frac{\partial J}{\partial \mathbf{U}_s^{n+1}}$ through geometry velocity function allows us to optimize through the geometry iso-surface.

### 2.3.4  Neural Fluid Control

We can train neural-network parameterized closed-loop fluid controllers with gradients fully computed at both geometry and velocity throughout time. We parameterize our controllers with a two-layer MLP. The controller takes as input the observation of the fluid velocity field at each frame and outputs dynamic control signals that affect the geometry through our parametric geometry presentation, which further affects the flow field. Our fully differentiable framework allows gradient-based methods to train the controller efficiently. We implemented the backbone of our code in C++ and CUDA for computational efficiency. We derived gradients for the geometry and simulation module analytically, then exposed the differentiable simulation framework through pybind11 [13] to enable seamless integration with deep learning libraries, which in our case is PyTorch [14].

## 3  Benchmarks and Applications

In this section, we introduce our fluidic design benchmarks and environments. We warp our environments using the standard protocol in the gym to facilitate learning practices. We present six fluidic design and control tasks (Fig. 2) to assess the effectiveness of our computational pipeline for fluidic

Table 1: **Task Specifications.** We summarize the simulation and optimization configuration for the design tasks shown in Sec. 3 and report the initial and optimized loss. We note that because our implementation adopts CFL condition for numerical stability during simulation, the actual steps simulated and back-propagated are higher than the numbers shown in "# Frames".

| | Resolution | # Frames | # Param. | $\frac{\partial L_f}{\partial \text{Design}}$ | $\frac{\partial L_f}{\partial \text{Control}}$ | Loss ($L_f$) Initial | Optimized |
|---|---|---|---|---|---|---|---|
| **Amplifier** | $64 \times 64$ | 40 | 5 | ✓ | | 13.401 | 0.005 |
| **Switch** | $64 \times 64$ | 120 | 10 | ✓ | ✓ | 13.162 | 1.893 |
| **Shape Identifier** | $128 \times 128$ | 10 | 10 | ✓ | | 70.152 | 0.759 |
| **Flow Modulator** | $40 \times 40 \times 40$ | 100 | 34 | ✓ | ✓ | 6.118 | 0.069 |
| **Neural Gate** | $40 \times 40 \times 40$ | 50 | 4.5k | | ✓ | 7.318 | 0.000 |
| **Neural Heart** | $48 \times 48 \times 48$ | 180 | 7.1k | ✓ | ✓ | 1.086 | 0.004 |

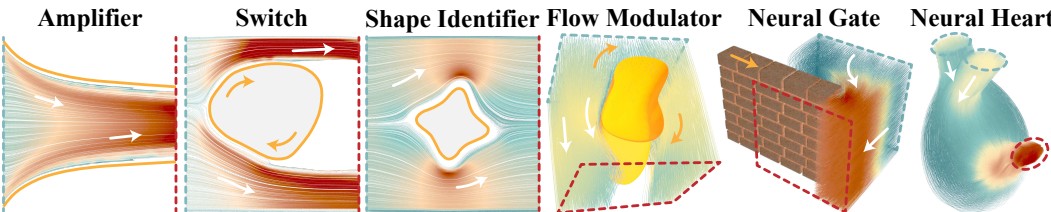

Figure 2: **Tasks Overview.** In each task, the blue dashed line represents the inlet, the red dashed line indicates the outlet, the white arrows show the flow direction, and the orange shapes and arrows denote the geometry and its motion direction.

system design and learning. A comprehensive illustration of these design scenarios, including the visualization of the optimization process, is provided in Appendix Sec C and our supplemental video. Initial conditions are set for all optimizations using randomly sampled values. We use Adam as our optimizer. We summarize the simulation configuration and optimization configuration as well as relevant statistics in Table 1.

## 3.1 Task Overview

**Amplifier** This design problem aims to amplify a parallel horizontal inflow by three times from an initial velocity of 5 units to 15 units. The device boundaries are parameterized as two symmetrically placed cubic Bezier curves. The design variables are the control points and endpoints of the two curves. The loss function is defined as the last frame L2 norm of the difference between the target and optimized fluid velocity norm. We visualize the initial and optimized designs in Fig. 3a. We overlay the design and the corresponding velocity field (colored by the norm) for both iterations.

**Shape Identifier** This task provides an example of system identification in a fluid environment by identifying the shape and position of a geometry, given observations of the flow field. We randomly

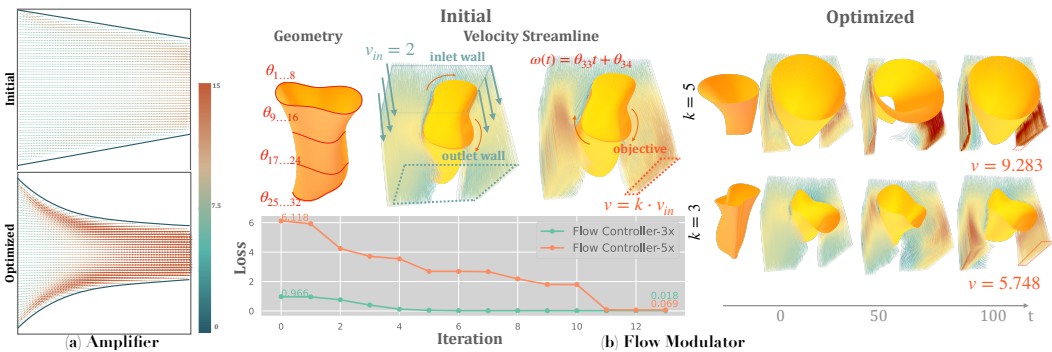

Figure 3: (a) Visualization of *Amplifier*. (b) Visualization of *Flow Modulator*.

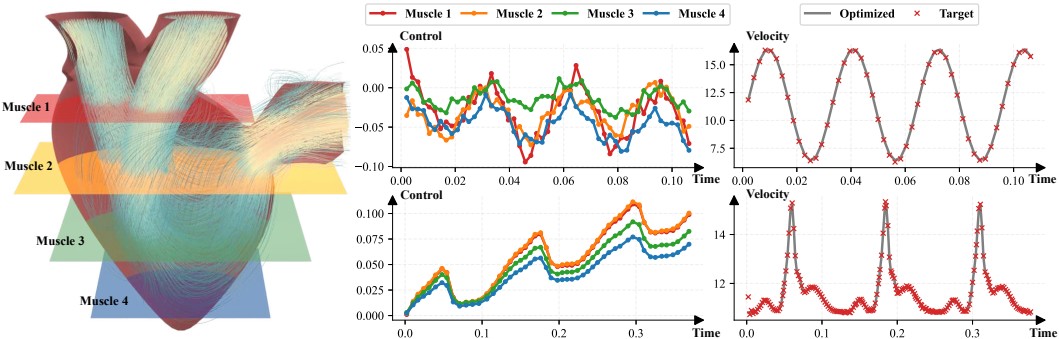

Figure 4: **Artificial Heart.** *Left*: visualization of the domain and the location of the muscles. *Middle*: Optimized control policy rollout visualization. *Right:* Optimization results visualization. The top and bottom diagrams visualize the cosine and the ECG target variants.

initialize the geometry in the domain. We define the loss function as the sum of the L2 norm of the velocity field difference to the observed ground-truth flow field across time. The optimization successfully reconstructs the shape and its position with random initialization (See Exp. 4.2).

**Switch**    This task simultaneously optimizes the geometry and the constant rotational speed of a 2D switch, allowing dynamic regulation of outlet flow velocity. A horizontal inflow at the left interacts with the switch, splitting into two distinct streams towards the right side. The goal is to let the time-dependent average velocity norm of the upper stream align with a predetermined flow profile. We define the loss function as the sum of the L2 norm of the difference between the average velocity norm of the fluid and the target velocity at each frame. The optimized design and control successfully generate a linearly increasing upper stream velocity norm profile, matching the specified target.

**Flow Modulator**    This task optimizes the geometry and control of a rotating 3D flow controller to achieve a target average outlet flow ($k\times$ inflow) at the domain's right boundary by the end of the simulation. The controller's geometry is parameterized by four 3D cross-sections, with rotation controlled by a sinusoidal function. We define the loss as the L2 norm of the difference between the average outlet and target velocity at the final frame. Fig. 3b illustrates the task: the top left shows initial parameters and task specification, bottom left shows optimization trajectories, and the right visualizes the optimized geometry and velocity streamline with two variants ($k = 3, 5$).

**Neural Gate Controller**    We learn a closed-loop controller for a 3D fluid gate moving horizontally. The controller is parameterized as a two-layer MLP. It observes the current outflow velocity through the gate and outputs the next frame motion offset to control the outflow velocity to match a target.

## 3.2    Scalability to Complex Fluid Fields: A Case Study of Artificial Heart

We showcase the scalability of our method through an artificial heart design and control learning task (Fig. 4). Artificial heart development is difficult due to the complex blood flow movement within the heart. This case study provides a first step in studying the heart's control strategies. We train a closed-loop controller that outputs the per-time-step contraction signal of the four muscles of a simplified heart model so that the outlet velocity matches a pre-defined target profile. The controller's states include temporal encoding of the current time step and the current outflow norm, and they are parameterized using a two-layer MLP. The heart's geometry is parameterized as the union of the two inlets, one outlet, and the heart chamber. In two variants of the task, one target flow profile is parameterized using a cosine curve (Fig 4 top), and one target flow profile mimics the shape of an electrocardiogram (Fig 4 bottom). We define the loss function as the sum of the L2 norm of the difference between the average velocity norm of the fluid and the target velocity at each frame. In both variants, the trained controller successfully outputs signals that generate blood flow that matches the target, demonstrating the effectiveness of our gradient-based optimization framework. We further visualize the rollouts of the trained controllers at Fig 4 left.

Table 2: **Time Performance.** Our method achieves one order of magnitude speedup across all resolutions compared to PhiFlow in both forward simulation and backward gradient propagation.

| Resolution | Forward | | | Backward | | |
|---|---|---|---|---|---|---|
| | PhiFlow (s) | Ours (s) | Speedup | PhiFlow (s) | Ours (s) | Speedup |
| $32 \times 32 \times 32$ | 1.282 | **0.024** | **53.4×** | 1.546 | **0.095** | **16.3×** |
| $40 \times 40 \times 40$ | 1.741 | **0.039** | **44.6×** | 1.983 | **0.121** | **16.4×** |
| $48 \times 48 \times 48$ | 2.227 | **0.068** | **32.8×** | 2.412 | **0.158** | **15.3×** |
| $64 \times 64 \times 64$ | 3.145 | **0.105** | **30.0×** | 4.094 | **0.301** | **13.6×** |

Table 3: **Memory and time performance comparison with DiffTaichi**

| Resolution | Memory (MB) | | Forward Time (s) | | Backward Time (s) | |
|---|---|---|---|---|---|---|
| | DiffTaichi | Ours | DiffTaichi | Ours | DiffTaichi | Ours |
| $32 \times 32 \times 32$ | 685 | **292** | 0.081 | **0.024** | 0.074 | **0.027** |
| $40 \times 40 \times 40$ | 1136 | **308** | 0.146 | **0.039** | 0.133 | **0.041** |
| $48 \times 48 \times 48$ | 1805 | **322** | 0.228 | **0.068** | 0.183 | **0.064** |
| $64 \times 64 \times 64$ | 5005 | **405** | 0.435 | **0.105** | 0.363 | **0.117** |

# 4 Experiments

## 4.1 Effects of Initialization on Optimization

This experiment studies the effect of random initialization in our fluid optimization tasks. Specifically, we conduct an experiment on the 3D heart controller task, which utilizes a neural network with 7,100 parameters. For this task, we initialized the network parameters with five different random seeds, tracking convergence under each condition. In the accompanying figure, we plot an extended version of the training trajectory, scaled logarithmically for better visualization, to compare the convergence of different random initializations. However, in practice, our method achieves the objective driven by the loss within tens of iterations, as is evident from the steep initial descent in the optimization curve. As shown in Fig. 5 left, the optimization consistently converges across all seeds, despite variations in initial network behavior. This consistency indicates the robustness of our method to random initialization even in high-dimensional optimization spaces, supporting its application to complex tasks in differentiable physics. These curves offer valuable guidelines for practitioners using our method in their deployments: the gradients from our approach are robust to hyper-parameters and scalable to high-dimensional optimization problems.

## 4.2 Gradient-Based vs Gradient-Free Optimization

We study the effectiveness of our gradient-based method against gradient-free optimization methods in the *Neural Heart* task (Fig. 5 right). We choose Proximal Policy Optimization (PPO) [15] as the baseline for reinforcement learning and CMA-ES [16] for evolution strategies. This environment is particular challenging due to the sensitivity of the bloodflow to the control signal changes across time, which could result in large flow field change if adjacent rollouts have large changes. We initialize all methods to output stochastic control signals of small noise for stable initial simulation. Our gradient-based method quickly converges to near zero after 60 epochs, while both gradient-free methods struggle in this environment. We argue that the rapid and successful convergence stems from the clear gradient provided by our method. Note that our differentiable optimization pipeline depends on the differentiability of the loss function (e.g., the imitation loss we used). This can be problematic if the objective is too complex to be characterized in a differentiable manner, in which case gradient-free methods are better alternatives. However, our framework will still excel due to its outstanding forward simulation speed, which we will elaborate next.

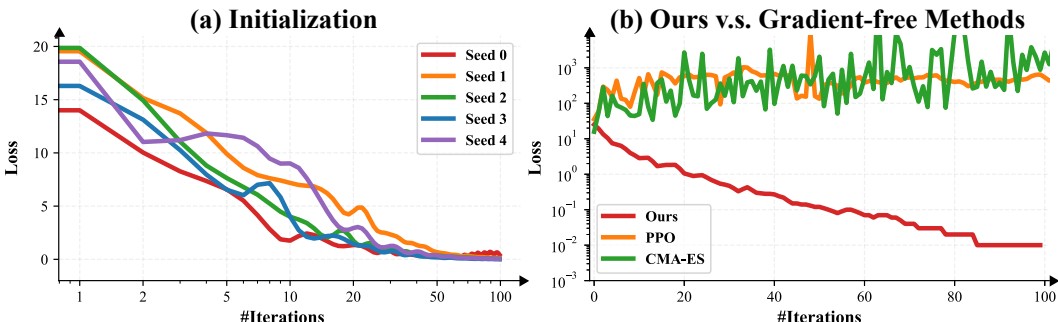

Figure 5: **Ablation Studies.** *Left:* Optimization trajectories for *Neural Heart* with 7100 parameters under different initialization. Iterations are visualized on a log scale. *Right:* Log scaled loss-iteration curves of our gradient-based method and other gradient-free optimization methods.

## 4.3 Time Performance Profiling and Comparison with PhiFlow

In this experiment, we demonstrate the performance efficiency of our framework through a comparison with PhiFlow. While PhiFlow operates with a TensorFlow-GPU backend, our framework is implemented in CUDA C++ and features a high-performance Geometric-Multigrid-Preconditioned-Conjugate-Gradient (MGPCG) Poisson solver [17]. To address the needs of differential operators and interpolations, which require access to neighboring cells in all directions, we divide the simulation domain into cubic blocks, each corresponding to a CUDA block. When launching a CUDA kernel, simulation data for each block is first loaded into shared memory, allowing efficient computation directly in shared memory and reducing global memory accesses. Additionally, to increase memory throughput, each block's data is stored consecutively in global memory. Our matrix-free MGPCG solver has a faster convergence rate than PhiFlow's Conjugate Gradient solver and uses a hierarchical grid data structure, with custom CUDA kernels for prolongation and restriction operations between coarse and fine grids.

We benchmark both the one-step forward simulation time and gradient back-propagation time at different resolutions, as shown in Table 2. The experiment runs on a workstation with an NVIDIA RTX A6000 GPU, where our framework consistently outperforms PhiFlow by an order of magnitude across all resolutions, benefiting both gradient-based and gradient-free optimization techniques. This performance improvement is particularly advantageous in fluid simulation applications, including robotics and video generation. Additionally, our system's gym protocol compatibility [4] makes it straightforward for practitioners to integrate and test our library. We plan to release our code and documentation upon acceptance.

## 4.4 Memory and Time Performance Profiling and Comparison with DiffTaichi

Here we compare our solver with DiffTaichi, a differentiable programming framework, to highlight the benefits of our approach in terms of scalability and efficiency. Our method, designed specifically for differentiable fluid simulation, uses manually derived gradients, avoiding the need to store intermediate computational graph at each timestep, unlike DiffTaichi, which relies on automatic differentiation. Additionally, our adjoint derivation for the projection solve step is independent of solver iterations, making our approach well-suited for advection-projection fluid simulations. To demonstrate this, we implemented a Conjugate Gradient (CG) solver for the projection step in DiffTaichi and compared time and memory performance across four grid resolutions in a 3D optimization scenario (Table 3). Our results show that our solver requires substantially less memory, with up to 12 times less memory usage than DiffTaichi at $64 \times 64 \times 64$ resolution. This reduction in memory stems from eliminating the need to store intermediate values during each CG iteration, making our solver particularly suitable for high-resolution, long-term optimizations.

## 5 Related Work

**Flow Control and Optimization** Beginning with the pioneering work of [18], a vast literature has been devoted to the optimization of fluid systems [19]. Given a predefined design domain with

boundary conditions, a typical optimization objective is to maximize some performance functional of a fluid system (e.g., the power loss of the system) constrained by the physical equations. Similar to a conventional structural optimization problem, the design domain is discretized. The optimization algorithm decides for each element whether it should be fluid or solid to optimize some performance functions such as power loss. Examples of flow optimization applications include Stokes flow [18, 20, 21, 22], steady-state flow [23], weakly compressible flow [24], unsteady flow [25], channel flow [26], ducted flow [27], viscous flow [28], fluid-structure interaction (FSI) [29, 30, 31], fluid-thermal interaction [32, 33], microfluidics [34], aeronautics [35, 36], and aerodynamics [37, 38], to name a few. [39] developed a dynamic differentiable fluid simulator and integrated the pipeline with neural networks for learning controllers. In computer graphics, [40] developed a differentiable framework to simulate and optimize flow systems governed by design specifications with different types of boundary conditions, while [8] developed an anisotropic material model to handle different boundary conditions using topology optimization framework. Both systems focus on the Stokes flow model and have not explored applications with a dynamic flow system. [41] adapted the adjoint method to control free-surface liquids.

**Differentiable Physics Simulation**    Differentiable simulations emerge and boost as a powerful tool to accommodate various optimization applications crossing graphics and robotics. A typical example is DiffTaichi [42], which created a differentiable programming environment to compute the gradients of physics simulations. A variety of physics simulation algorithms stemming from graphical applications have been adapted to a differentiable framework to facilitate inverse design applications, including fluids [43, 44, 45], position-based dynamics [7], cloth [46, 47], deformable objects [48], articulated bodies [49], object control [50], and solid-fluid coupling systems [39, 51]. While [51] proposed a method to differentiate Lagrangian fluid simulation, optimization of rigid geometry is not discussed. Many applications across graphics and robotics have been explored, such as soft-body design and locomotion [52, 53] and fluid manipulation [5]. However, none of these approaches focused on enabling the inverse design of fluidic device systems in dynamic Navier-Stokes flow.

**Computational Design**    The last decade has witnessed an increasing interest in the design of computational tools and algorithms targeting the digital fabrication of physical systems. A broad range of applications have been addressed, including the mechanical characters [54, 55, 56], inflatable thin shells [57], foldable structures [58, 59], Voronoi structures [60], joints and puzzles [61], spinning objects [62], buoyancy [63], gliders [64], multicopters [65], hydraulic walkers [66], origami robots [67], articulated robots [68], and multi-material jumpers [69], to name just a few. Among these applications, the problem of optimizing the shape and control of a 3D printable object to manifest specific mechanical properties and functionalities has drawn particular attention. Examples of designing mechanical properties by optimizing materials include optics [70, 71], mechanical stability [72], strength [73], rest shape [74], and desired deformation [75, 76].

## 6    Conclusions, Limitation and Future Work

In this paper, we proposed a fully differentiable pipeline for neural fluidic system control and design, addressing the challenges of complex geometry representation, differentiable fluid simulation, and co-design optimization processes. Our pipeline features a low-dimensional parametric geometry representation and a differentiable Navier-Stokes simulator for predicting fluid behavior. We demonstrate the effectiveness of our pipeline in a number of complex control design tasks, ranging from different fluidic functional controls to complex neural heart control.

There are certain limitations and avenues for future work. First, the current pipeline assumes the standard Navier-Stokes model, which limits its applicability to Newtonian flow. Extending the framework to handle non-Newtonian flows or multi-physics interactions would be an interesting direction for future research. Additionally, the pipeline relies on parametric representation, which may encounter challenges in navigating complex design spaces with high-dimensional or discontinuous parameterizations such as coupling control design with topology optimization. Exploring alternative optimization algorithms or incorporating surrogate models could enhance the efficiency and robustness of the optimization process. Furthermore, while we demonstrate the effectiveness of our pipeline in several control and design tasks, additional validation and bench-marking against real-world physical experiments would be valuable to establish the pipeline's reliability and generalizability.

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

# A  Geometry Representation Implementation

**2D Geometry**  We define a closed 2D surface using $N$ connected cubic Bezier curves parameterized in the polar coordinate frame. A point $\mathbf{c}$ is defined on the surface to establish the center of a polar coordinate frame. Each Bezier curve spans an arc of $\frac{2\pi}{N}$ radians on the polar coordinate plane, and its control points are symmetrically placed, each at an angular displacement of $\frac{2\pi}{3N}$ radians from their corresponding curve endpoint. The shape of each Bezier curve, and consequently the overall surface, is manipulated via two scalar parameters, $\rho_1$ and $\rho_2$, dictating the polar coordinate distance of the two control points. The $i$-th cubic Bezier curve is defined by two control points $\mathbf{p}_0 = (\rho_1^i \cos\theta, \rho_1^i sin\theta) + \mathbf{c}$ and $\mathbf{p}_1 = (\rho_2^i \cos\theta, \rho_2^i sin\theta) + \mathbf{c}$, where $\mathbf{c}$ is the reference center point. The endpoints $\mathbf{e}_0$, $\mathbf{e}_1$ are computed by ensuring $\mathbf{e}_1^i = \mathbf{e}_0^{(i+1)\%N}$ and colinearity of each pair of $\mathbf{p}_1^i, \mathbf{e}_1^i, \mathbf{p}_0^{(i+1)\%N}$. This representation offers a compact way of defining diverse geometries.

**3D Geometry**  We parameterize a closed 3D surface using 2D surfaces defining the key cross-sections of the geometry along the $z$-axis of the local object frame, where each 2D surface is parameterized by N cubic Bezier curves. The parametrization includes $z = z_0$ and $z_1$, which determines the Z plane of the first and last cross-section and the parameters for each key cross-section. The $i - th$ key cross-section is defined by the center $\mathbf{c}^i$ and control point parameters $\rho_1^i, \rho_2^i$ for each of $i \in [1, 2, \ldots, N]$. The key cross-sections are assumed to be evenly spaced along the z-axis. Then, given $z_0 \leq z \leq z_1$, the cross-section of the closed surface at $Z = z$ is defined by interpolating the centers and control points of all key cross-sections using the interpolation scheme

$$
\begin{cases}
\mathbf{c}_z = \sum_{i=1}^{n} \mathbf{c}^i \left(\frac{z - z_0}{z_1 - z_0}\right)^i & \text{(10a)} \\[2ex]
\rho_j = \sum_{i=1}^{n} \rho_j^i \left(\frac{z - z_0}{z_1 - z_0}\right)^i & \text{(10b)}
\end{cases}
$$

# B  Temporal Discretization of the Governing Equation

We build the fluid simulator by leveraging the operator-splitting method [9][10]. Each simulation step comprises of advection, viscosity, and projection.

**Advection**  We employ the semi-Lagrangian advection scheme specified in (Eqs. 11 and 12) to propagate velocity through the fluid domain:

$$\frac{\tilde{\boldsymbol{u}}^{n+1/2} - \boldsymbol{u}^n}{\Delta t/2} = -\boldsymbol{u}^n \cdot \nabla \boldsymbol{u}^n, \tag{11}$$

$$\frac{\tilde{\boldsymbol{u}}^{n+1} - \boldsymbol{u}^n}{\Delta t} = -\tilde{\boldsymbol{u}}^{n+1/2} \cdot \nabla \boldsymbol{u}^n. \tag{12}$$

**Viscosity** For incompressible fluid with a constant viscosity coefficient, the viscous force density is equivalent to the product of the Laplacian of velocity and the viscosity coefficient. We employ explicit time integration to update the fluid velocity in response to the viscous force.

$$\frac{\hat{\boldsymbol{u}}^{n+1} - \tilde{\boldsymbol{u}}^{n+1}}{\Delta t} = \nu \nabla^2 \tilde{\boldsymbol{u}}^{n+1} \tag{13}$$

**Projection** The projection step involves an update of the pressure and the velocity field (Eq. 14a) to ensure the satisfaction of the incompressibility condition (Eq. 14b).

$$\begin{cases} \dfrac{\boldsymbol{u}^{n+1} - \hat{\boldsymbol{u}}^{n+1}}{\Delta t} = -\dfrac{1}{\rho} \nabla p^{n+1}, & \text{(14a)} \\ \nabla \cdot \boldsymbol{u}^{n+1} = 0. & \text{(14b)} \end{cases}$$

On boundaries, the pressure is regulated by two conditions: the Dirichlet boundary condition (Eq. 15a) and the non-penetrating Neumann boundary condition (Eq. 15b) given computed geometry velocity $\boldsymbol{u}_s^{n+1}$ :

$$\begin{cases} p^{n+1} = 0, & \boldsymbol{x} \in \partial \Omega_f^{n+1}, & \text{(15a)} \\ \boldsymbol{u}^{n+1} \cdot \boldsymbol{n} = \boldsymbol{u}_s^{n+1} \cdot \boldsymbol{n}, & \boldsymbol{x} \in \partial \Omega_b^{n+1}. & \text{(15b)} \end{cases}$$

The pressure field for the subsequent time step $p^{n+1}$ is determined by solving the resultant Poisson equation (Eq. 16).

$$\frac{\Delta t}{\rho} \nabla^2 p^{n+1} = \nabla \cdot \hat{\boldsymbol{u}}^{n+1}. \tag{16}$$

## C  Additional Optimization Task Details and Visualization

### C.1  Switch

We visualize the initial and optimized design for the amplifier task in Fig. 6.

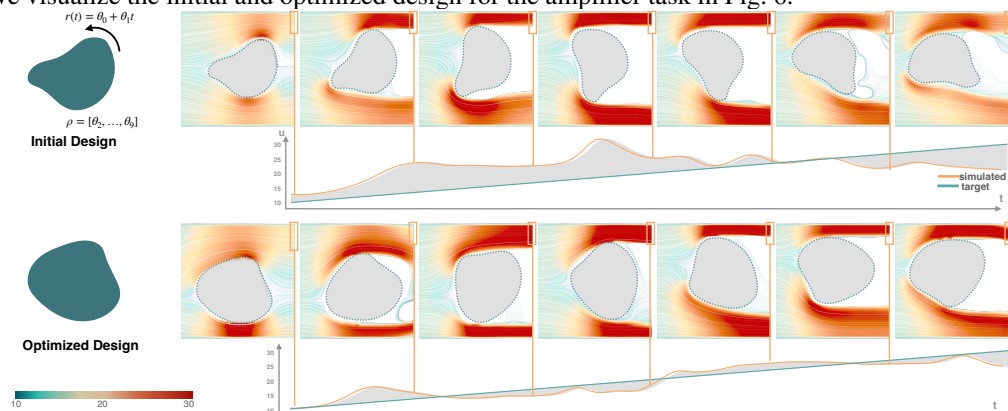

Figure 6: Fluidic Switch. The switch rotates dynamically across time (dotted lines). The shape of the switch is parameterized as a 2D Polar Bezier, whose parameters, along with the parameters of the rotation signal are subject to optimization. The top and bottom of the illustration visualize information from the initial and optimized iteration respectively. For each iteration, we visualize the design geometry (left) and corresponding streamlines of the flow field (right) at 7 key-frames evenly sampled across time. We additionally plot the target (green) and outlet velocity norm profile (orange) across time and visualize their difference in grey shaded area.

## D  Experiment on Fluid Solver Validation – Kármán Vortex Street

To further validate the performance of our solver, we conducted an additional experiment simulating the formation of a classic **Kármán Vortex Street**. This experiment was executed at a resolution of $512 \times 1024$ and illustrates the capability of our solver in capturing complex fluid dynamics phenomena. As shown in Fig. 7, we simulate a horizontal flow passing around a cylindrical obstacle

at three distinct kinematic viscosity values: inviscid ($\nu = 0$), moderate viscosity ($\nu = 0.01$), and high viscosity ($\nu = 0.1$).

Each viscosity setting demonstrates the characteristic vortex shedding pattern associated with Kármán Vortex Street formation. These results confirm our solver's ability to replicate this well-known phenomenon and offer insights into the effect of varying viscosity on vortex behavior. This validation experiment supports the accuracy and versatility of the solver across different fluid conditions.

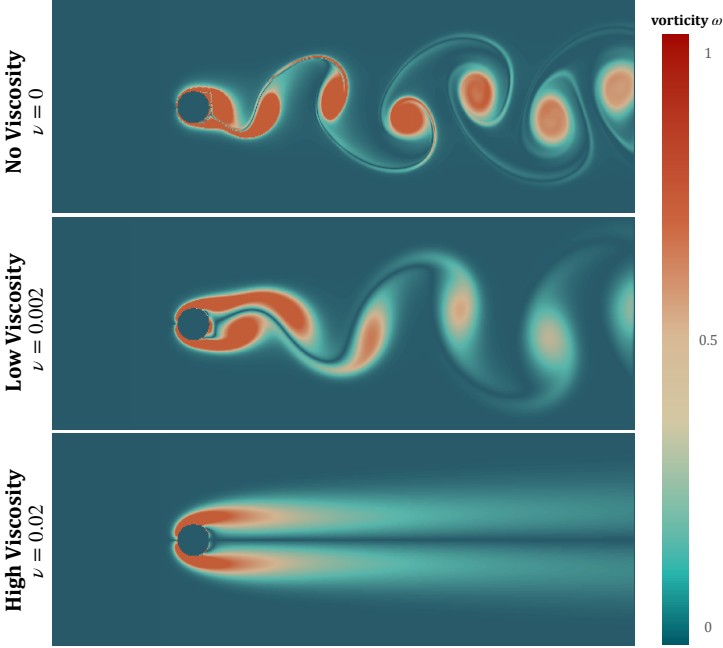

Figure 7: Solver Validation. Visualization of Karman Vortex Street under different viscosity conditions. Here, we illustrate the results of the classic Karman Vortex Street test for three different kinematic viscosity values (From top to down $\nu = 0.0$, $\nu = 0.002$ and $\nu = 0.02$), simulated using our differentiable simulator within a domain size $512 \times 1024$. Each figure visualizes the vortex patterns, and the results demonstrate how increased viscosity leads to a notable change in vortex formation and dissipation.

## E   Gradient Stability and Solver Steps Statistics

Gradient stability in differentiable physics is a well-known challenge, particularly given the potential for gradient explosion or vanishing when gradients are accumulated across numerous solver steps. In our approach, however, we have not encountered significant issues with gradient stability. This stability is likely due to the accuracy of the gradients produced by our framework and the robustness of our numerical solver. To illustrate this, we provide gradient norm statistics over the full course of optimization for three tasks of varying complexity in Table 4, demonstrating consistent gradient magnitudes without evidence of explosion or vanishing. For additional robustness, our implementation includes gradient clipping with a threshold of 1.0, which can mitigate gradient explosion in particularly challenging scenarios. This technique ensures gradients remain within manageable limits and contributes to the overall stability of our optimization pipeline.

Furthermore, the actual number of solver steps required to advance between frames in our solver depends on the Courant-Friedrichs-Lewy (CFL) condition, which is maintained to ensure numerical stability. Additional statistics on solver steps over one optimization cycle across various tasks are provided in the upper portion of Table 4, offering further insight into the computational demands and stability characteristics of our approach.

| Metric | Statistic | Shape Identifier | Heart 3D | Gate 3D |
|---|---|---|---|---|
| **Steps** | Mean, Std | (11, 0) | (54, 0) | (58, 0) |
| **Gradient Norm** | Min, Max | (0.46, 252.37) | (1.61, 820.82) | (5.13, 169.21) |
| | Mean, Std | (28.04, 38.00) | (65.96, 109.40) | (40.71, 34.36) |

Table 4: Statistics of the gradient norm and step count over the full course of optimization for three tasks of varying complexity.

Table 5: Gradient Validation for Shape Identifier Task

| | | **Gradient Values by Parameter (P1 to P6)** | | | | | |
|---|---|---|---|---|---|---|---|
| | | **P1** | **P2** | **P3** | **P4** | **P5** | **P6** |
| **Analytic** | | 0.048 | 0.394 | 0.314 | 0.046 | 0.133 | 0.067 |
| **Finite Diff** | | 0.048 | 0.396 | 0.314 | 0.046 | 0.134 | 0.066 |
| **Abs Diff** | | -1.7e-4 | -2.1e-3 | 5.2e-4 | -3.4e-4 | -1.4e-3 | 7.2e-4 |
| **Elem Err** | | 0.004 | 0.005 | 0.002 | 0.007 | 0.011 | 0.011 |
| | | **Gradient Values by Parameter (P7 to P11)** | | | | | |
| | | **P7** | **P8** | **P9** | **P10** | **P11** | |
| **Analytic** | | 0.087 | 0.008 | -0.022 | 0.540 | -0.058 | |
| **Finite Diff** | | 0.086 | 0.008 | -0.025 | 0.540 | -0.059 | |
| **Abs Diff** | | 2.1e-4 | -9.6e-5 | 2.2e-3 | 2.4e-4 | 9.5e-4 | |
| **Elem Err** | | 0.003 | 0.012 | 0.098 | 0.000 | 0.016 | |

## F  Experiment on Gradient Validation

To ensure the correctness of the gradients in our differentiable simulation framework, we validated the analytical gradients of all kernels, functions, and the entire simulation and optimization pipeline using finite difference approximations. Specifically, we employed the central difference method with a step size of $1.2 \times 10^{-5}$ to approximate the gradients numerically and compared them with the analytical gradients calculated by our solver.

In this validation experiment, we consider the end-to-end gradients for the Shape Identifier Task, which involves optimizing over 11 parameters. The analytical gradients, finite difference gradients, their absolute differences, and element-wise errors are reported in Table 5. For this task, we observed a relative error of 0.0047 for the gradient vector, confirming the high accuracy of our analytical gradients.

