# OpenReview forum: "NeuralFluid: Nueral Fluidic System Design and Control with Differentiable Simulation"
_NeurIPS.cc/2024/Conference — NeurIPS 2024 poster_

### Official Review · Reviewer_ueL6 · 2024-06-29

**Soundness:** 3
**Presentation:** 2
**Contribution:** 3
**Rating:** 5
**Confidence:** 3

**Summary:**

The paper details a framework for design and control of complex fluidic systems with soft and hard boundaries. The paper specifically develops a fully differentiable pipeline that is able to optimize the design of the mesh to achieve the specified fluid control goal. The proficiency of the proposed framework has been demonstrated on multiple design and fluid control tasks like: (i) amplifying flow of a fluid by changing the shape of an initial boundary (ii) optimizing the position of a "neural gate" to satisfy the target outflow requirement (iii) designing an artificial heart (starting from an initial parameterized representation) to achieve a low-error match w.r.t a target simulation. The diversity of the tasks as well as task complexity and good results make the contributions in the paper worth considering. However, the paper lacks crucial details without which it is hard to fully appreciate the effectiveness of the proposed methods, hence making it hard to support publication of the paper in the present state.

**Strengths:**

* The paper tackles an interesting, challenging and important problem of developing a (fully differentiable) framework for optimal design and control of fluid flows in challenging contexts.


* The results presented in the paper have been demonstrated on challenging tasks (i.e., not the usual "toy" problems tackled by other such papers) and as presented seem to be effective on the design and control tasks investigated.

**Weaknesses:**

* Although the results seem interesting and the problem the paper tackles is challenging, the paper lacks articulation of crucial details making the extent of contribution hard to appreciate. Some crucial aspects of the paper need further clarification.

> Q1. How is the "initial parametric geometry" (required as input by the framework) generated? What are the feasibility constraints as far as this initial geometry is concerned? Where can a new user of this framework obtain such an initial geometry to supply to the system? An analysis of this or at least an articulation for usability by users is necessary.

> Q2. Further details of the experiment setup are necessary. For example: what is the design of the PPO, CMA-ES models employed? What is the reward function for PPO and what were the architectures of the learnable components therein?

> Q3. Why was `PhiFlow` chosen as the framework for comparison? Wouldn't `DiffTaichi`[1] be a more apples-to-apples comparison for measuring speedup, considering that the C++ implementation of underlying libraries similar to the current proposed framework?

* Section 2.2  needs further detailing for a machine learning audience who may not be familiar with `Constructive Solid Geometry`.

* Code pipeline has not been made available for review. Hence, the results are currently not reproducible. It is imperative to make code available during the review stage.

### `References`
1. Hu Y, Anderson L, Li TM, Sun Q, Carr N, Ragan-Kelley J, Durand F. Difftaichi: Differentiable programming for physical simulation. arXiv preprint arXiv:1910.00935. 2019 Oct 1.

**Questions:**

See questions in `Weaknesses` section.

**Limitations:**

One improvement that can be made would be to highlight the details of the initial design specifications for the various tasks and enumerating some general prescriptions about generating good initial designs to be supplied to the pipeline.

---

> ### Author Rebuttal · Authors · 2024-08-07
>
> We thank the reviewer for the careful review and insightful questions.
>
> **1. How is the "initial parametric geometry" generated? What are the feasibility constraints...?**
> * Thank you for bringing up this important question. Usability is crucial for allowing users to specifying complex shapes enabled by our framework. We therefore designed the geometry specification process similar to existing CAD software.
> * For instance, a 3D component can be created by extruding a 2D shape along an axis, and users can union or intersect 3D models to form complex structures. Each 2D component is defined using a polar Bézier specification. This approach ensures that users with basic CAD software experience can effectively use our model.
> * For example, the heart model presented in our paper was created by specifying each heart component using six cross-sections, each made of circles with various radii and center positions. This method makes the geometry specification process intuitive and accessible. * By leveraging familiar CAD concepts, we aim to make our framework user-friendly and accessible to researchers and practitioners. We will also provide detailed documentation, visualization tools, and examples to further assist new users in generating the initial parametric geometry needed for their specific applications when releasing the codebase.
>
> **2. Code pipeline has not been made available for review.**
>
> We will open-source our code upon acceptance. We have provided the anonymized version of our code at https://anonymous.4open.science/r/DiffFluidRebuttal-1D24/ for the review stage.
>
> **3. Sec 2.2 needs further detailing for a ML audience who may not be familiar with CSG.**
>
> Thank you for the suggestion. We will add a more detailed description of CSG in our revised manuscript. Briefly speaking, CSG defines a hierarchical tree structure to represent volumetric geometry. Each leaf node in the tree denotes a geometric primitive (e.g., the Bezier surfaces in Sec. 2.2), and each intermediate node up to the root forms a new volumetric geometry by applying boolean operators to its child nodes, e.g., taking the union of the volumetric geometries represented by the child nodes. The root node represents the final geometry, which is formed by recursively applying boolean operators (typically unions, intersections, and subtractions) to a set of geometric primitives stored in the leaf nodes.
>
> **4. Further details of the experiment setup are necessary. What is the design of the PPO, CMA-ES models employed? What is the reward function for PPO and what were the architectures of the learnable components therein?**
>
> Our PPO implementation is based on the open-sourced code of pytorch-a2c-ppo-acktr. Specifically, the actor and critic networks are both two-layer MLPs with tanh activations and a hidden layer of 64 neurons. For both controller tasks, we set the observation to be the same as the input to our gradient-based closed-loop controller. The reward function is the negative comparative of the loss function of the gradient-based closed-loop controller, plus a positive component to avoid negative rewards. Our CMA-ES implementation is based on the open-source Nevergrad. We set the population size to 10 and the metric to be the same as the loss function used for our method. We will include the details of the baseline in the Appendix.
>
> **5. Why was PhiFlow chosen as the framework for comparison? Wouldn't DiffTaichi[1] be a more apples-to-apples comparison for measuring speedup, considering that the C++ implementation of underlying libraries similar to the current proposed framework?**
>
> *Thank you for your valuable feedback. We chose PhiFlow for our comparison because it is an open-source, differentiable simulator specifically optimized for fluid simulation, which aligns closely with our objectives. While DiffTaichi is indeed a powerful differentiable programming tool, it serves a broader range of applications similar to Warp or JAX.
> * However, we recognize the importance of providing a comprehensive comparison for practitioners. We have conducted additional experiments to include a comparison with DiffTaichi. In these experiments, we implemented a Conjugate Gradient (CG) iterative solver for the projection step, which is the most time-consuming component of each fluid solve, in DiffTaichi and compared its time and memory performance with our solver in 3D under 4 resolutions. The results of this comparison are detailed in Table 2.
> * To summarize, our implementation is approximately 3-4 times faster than DiffTaichi in both forward and back propagation. This performance gain is likely due to our efficient CUDA kernel tailored for the Laplacian operator and the rapid convergence of our MGPCG solver. Additionally, our solver uses significantly less memory than DiffTaichi, with the advantage becoming more pronounced at higher grid resolutions. At 64x64x64, DiffTaichi uses 12 times more memory. This is because DiffTaichi relies on automatic differentiation and needs to store the computational graph and values for all CG iterations, whereas our adjoint gradient method does not require storing these values. This makes our solver more suitable for long-term optimization at high resolutions, where efficient memory usage is crucial.
>
> **6. One improvement that can be made would be to highlight the details of the initial design specifications for the various tasks and enumerating some general prescriptions about generating good initial designs...**
>
> Thank you for your valuable suggestion. For training fluid controllers, such as in the neural gate and neural heart tasks, we have found that initializing the network to output small actions helps achieve stable initial simulations and facilitates faster convergence of the network. In our implementation, this is accomplished by initializing all network layers with orthogonal matrices scaled to a small norm. We will include additional details on initial design in the Appendix.

---

> ### Author Response · Authors · 2024-08-12
>
> Dear reviewer ueL6, we wanted to check if there are any remaining questions or concerns about our rebuttal. Since the author-reviewer discussion period ends tomorrow, we’d appreciate any final feedback you might have.
>
> Thank you for your time and attention.

---

> ### Author Response · Authors · 2024-08-13
>
> Dear Reviewer,
>
> As the discussion deadline approaches, we are eager to address any remaining concerns you may have. We kindly request you to review our responses. If there is anything further we can do to facilitate the evaluation process, please let us know. We also respectfully ask that you reconsider our score based on the rebuttal provided.
>
> Thank you for your attention.
>
> Best regards,
>
> Authors

---

> > ### Comment · Reviewer_ueL6 · 2024-08-13
> > **Official Comment by Reviewer ueL6**
> >
> > The author responses along with sharing of the anonymous link has enabled me to inspect the code. I am satisfied regarding the code. However, I still have concerns regarding the sensitivity of the method to the `initial parametric geometry`. But based on the additional results about `DiffTaichi` and the sharing of the code, I am able to raise my score from 4 --> 5.

---

> > > ### Author Response · Authors · 2024-08-13
> > >
> > > Dear Reviewer,
> > >
> > > Thank you for taking the time to review our work and for raising your score.
> > >
> > > Regarding your concern about the sensitivity of the method to the initial parametric geometry, we acknowledge the importance of this aspect. In our original paper, Section 4.1, we conducted an experiment on the Effects of Initialization on Optimization. Specifically, in the Shape Identifier task, which aims to identify the shape and center location of geometry parameterized by 10 parameters, we initialized geometries of various shapes with different center locations in space using 5 different random seeds. Despite the varying initial losses (ranging from 27 to 59), the optimization process proceeded smoothly, and all cases reached a final loss near zero.
> > >
> > > Additionally, in our rebuttal, we extended this analysis to more complex tasks. In Figure 2 of the rebuttal PDF, we presented an additional verification using the 3D heart controller task, which involves 7.1k neural network parameters subject to optimization. We initialized the network parameters with 5 different random seeds, resulting in different initial actuations and geometries. Despite these variations, we observed consistent convergence across all seeds, even with different initial behaviors, further demonstrating the robustness of our method.
> > >
> > > We appreciate your feedback and please let us know if you have further comments or questions.

---

### Official Review · Reviewer_QmuX · 2024-07-08

**Soundness:** 2
**Presentation:** 2
**Contribution:** 2
**Rating:** 5
**Confidence:** 4

**Summary:**

This paper introduces a novel approach to fluidic system design and control using differentiable simulation. The authors propose a method that leverages gradient-based optimization to enhance performance and accuracy in fluid dynamics applications.

**Strengths:**

The paper demonstrates the effectiveness of their approach by comparing it to gradient-free methods such as Proximal Policy Optimization (PPO) and Covariance Matrix Adaptation Evolution Strategy (CMA-ES).
The authors effectively highlight the advantages of having access to gradients, allowing for better optimization due to more information about the underlying function. In contrast, gradient-free methods approximate these gradients, which can be less efficient.
The approach is particularly beneficial in scenarios where the underlying dynamics are smooth, as opposed to scenarios better suited for gradient-free methods.

**Weaknesses:**

The concept of backpropagation through time in reverse order has been extensively explored in neural ordinary differential equations (neural ODEs) and various adjoint methods literature.
The paper appears to be closely related to existing work but utilizes a different discretization or numerical method. The claim of differentiability may be redundant as the equations themselves, or their adjoint operators, are inherently differentiable.
The approach resembles the process of writing custom Jacobian-vector products (JVPs), which may not significantly advance the field.Although the authors outline the limitations of their method for other fluid models, they fail to address significant limitations encountered during backpropagation, such as the need to checkpoint the solution or the increasing memory use as a function of simulation length.

**Questions:**

The majority of the novelty comes from an efficient CUDA kernel implementation but the specifics are omitted.
What specific design considerations are taken that enable this method to outperform existing solvers?

**Limitations:**

Although the authors outline the limitations of their method for other fluid models, they fail to address significant limitations encountered in many differentiable fluid simulations. Specifically,during backpropagation, such as the need to efficiently checkpoint the solution or the dependence of the amount of memory needed.

---

> ### Author Rebuttal · Authors · 2024-08-07
>
> We are grateful for the reviewer’s insightful comments and questions.
>
> **1. The majority of the novelty comes from an efficient CUDA kernel implementation but the specifics are omitted. What specific design considerations are taken that enable this method to outperform existing solvers?**
> * Because differential operators and interpolations access neighboring cells of a spatial point in all directions, we divide the whole simulation domain into cubic blocks, and each cubic block corresponds to a CUDA block. When launching a CUDA kernel, we first load the simulation data of a block into shared memory, and perform calculation efficiently on shared memory. To increase the memory throughput of the data transfer between global memory and shared memory, the simulation data of a cubic block is stored consecutively on global memory.
> * While we adopt a matrix-free MGPCG solver for Poisson, Phiflow uses a CG solver. Therefore, our solver has a much better convergence rate than theirs. To implement an efficient MGPCG solver, we also devised a hierarchical grid data structure on GPUs and customized CUDA kernels for prolongation and restriction operations between coarse and fine grid. Our GPU solver is highly optimized and specially tuned for fluid simulation tasks, which will be open-source.
>
> **2. ...such as the need to efficiently checkpoint the solution or the dependence of the amount of memory needed.**
>
> * We are glad that the reviewer raised this limitation. Gradient checkpointing is a common technique in deep learning applications involving large models. However, our method is specifically designed for differentiable fluid simulation, where all the gradients have been manually derived for efficiency instead of relying on automatic differentiation like DiffTaichi. As a result, our method does not require the automatic differentiation workflow to store all the intermediate activations inside each timestep. In particular, our adjoint derivation of the project solve step is independent of the number of iterations in the solver, which makes our method a way better choice for advection-projection fluid simulation than any existing baselines.
> * To support our claim, we have conducted additional experiments to include a comparison with DiffTaichi, which uses automatic differentiation to derive gradients. We implemented a Conjugate Gradient (CG) iterative solver for the projection step, which is the most time-consuming component of each fluid solve, in DiffTaichi and compared its time and memory performance with our projection solver across four different resolutions in a 3D optimization scenario. The results of this comparison are detailed in Table 2.
> * To summarize, our solver uses significantly less memory than DiffTaichi, with the advantage becoming more pronounced at higher grid resolutions. For instance, at a 64x64x64 resolution, DiffTaichi uses 12 times more memory. This is because DiffTaichi relies on automatic differentiation and needs to store the computational graph and values for all CG iterations, whereas our adjoint gradient method does not require storing these values. This makes our solver more suitable for long-term optimization at high resolutions, where efficient memory usage is crucial. In conclusion, we acknowledge that gradient checkpoint is necessary when the computational graph grows, but our method has optimized the memory consumption to the best extent.

---

> > ### Comment · Reviewer_QmuX · 2024-08-13
> >
> > Thank you to the authors for their detailed response.
> >
> > Based on the additional information provided, it seems that the primary contribution lies in the custom fluid solver implementation. However, the authors have not sufficiently clarified the novelty of their approach beyond the solver itself or its broader impact on the NeuRIPS community. That being said, the paper is stronger given the additional details. I will revise my score to 4 --> 5.

---

> ### Author Response · Authors · 2024-08-12
>
> Dear reviewer QmuX, we wanted to check if there are any remaining questions or concerns about our rebuttal. Since the author-reviewer discussion period ends tomorrow, we’d appreciate any final feedback you might have.
>
> Thank you for your time and attention.

---

> ### Author Response · Authors · 2024-08-13
>
> Dear Reviewer,
>
> As the discussion deadline approaches, we are eager to address any remaining concerns you may have. We kindly request you to review our responses. If there is anything further we can do to facilitate the evaluation process, please let us know. We also respectfully ask that you reconsider our score based on the rebuttal provided.
>
> Thank you for your attention.
>
> Best regards,
>
> Authors

---

### Official Review · Reviewer_ZAVx · 2024-07-12

**Soundness:** 4
**Presentation:** 4
**Contribution:** 4
**Rating:** 8
**Confidence:** 3

**Summary:**

The paper proposes a new set of utilities for experimenting with system design and control of viscous fluid flows on deformable domains. The contributions include (a) a differentiable NSE solver, (b) Bezier curve-based geometry parametrization, (c) an algorithm to jointly optimize a control and design objective, (d) a reinforcement learning environment interface, and (e) benchmark cases with baseline results. The results demonstrate the superiority of the differentiable solver framework over existing gradient-free approaches and pave the road for exciting research, e.g., in medical applications.

**Strengths:**

- Implementing an efficient PDE solver from scratch in C++ and CUDA and then analytically deriving gradients is a great service to the community.
- The manuscript is well written and provides a good balance between theory and experimental evidence.
- Efficiently parametrizing complex geometries has a great potential for bridging the gap between toy experiments and real-world problems.

**Weaknesses:**

- **A. Proper figures**: make the figures on pages 3 and 4 proper figures with captions. Currently, the only way to refer to them is as "the figure on paper 3 ..."
- **B. Solver validation**: having some experience implementing numerical solvers, I know that validating the implementation is crucial. Could you add an appendix on validating the solver on an established fluid mechanics benchmark, like channel flow? The same applies to the gradients, which are typically validated by comparing them against a finite difference numerical approximation.
- **C. Minor issues**: Line 131 "can derived"?; Line 155 "of from"?

**Questions:**

- Did you consider submitting to the Datasets & Benchmarks track instead of the general conference? Your contribution feels more like a benchmarking suite than machine learning research.
- In the caption of Table 1, you mention that "# Frames" is not the same as the number of solver steps. What is the actual number of solver steps between two frames? Accumulating gradients over hundreds of steps often leads to exploding gradients, so why don't you have these issues if you stick to the CFL number?
- Related to the previous question, using learned surrogates as PDE solvers has demonstrated great potential in overcoming the exploding gradients issue by simply learning to do as much as 200x larger time integration steps [1]. Does your framework allow for replacing the NSE solver with a learned surrogate, e.g. a U-Net or FNO?

---
[1] Allen et al., "Inverse Design for Fluid-Structure Interactions using Graph Network Simulators", NeurIPS 2022

**Limitations:**

The authors have fairly assessed the limitations of the proposed approach.

---

> ### Author Rebuttal · Authors · 2024-08-07
>
> We thank the reviewer for appreciating our work and raising constructive suggestions.
>
> **1. Proper figures: make the figures on pages 3 and 4 proper figures with captions"**
>  We will make both inset figures proper figures in the revision.
>
> **2. Solver validation: having some experience implementing numerical solvers, I know that validating the implementation is crucial. Could you add an appendix on validating the solver on an established fluid mechanics benchmark, like channel flow?**
>
> Thank you for your suggestion. We have added an experiment validating our solver using the classic Karman Vortex Street under a resolution of 512x1024. In Fig.1, we show the vortex pattern formed by a horizontal flow passing through a cylinder at three different kinematic viscosity values: $\nu = 0.0$ (inviscid fluid), $\nu = 0.002$, and $\nu = 0.02$ to recreate the classic phenomenon. We will add this experiment to the Appendix.
>
>
> **3. The same applies to the gradients, which are typically validated by comparing them against a finite difference numerical approximation.**
>
> Indeed, it is important to validate the correctness of the gradients. During the development of our differentiable simulation, we validated the correctness of our analytical gradients for all kernels, functions, and the simulation and optimization pipeline using finite difference approximations. Below, we include the validation experiment for the end-to-end gradients of the Shape Identifier Task, which has a total of 11 optimization parameters. We compute the finite difference gradient using the central difference with a step size of 1.2e-5. Below we report the two gradients, their element-wise, as well as the vector error and difference.
>
> |||||||||||||
> |---|---|---|---|---|---|---|---|---|---|---|---|
> |Analytical|0.0483|0.3935|0.3142|0.0459|0.1327|0.0670|0.0866|0.0081|-0.0224|0.5404|-0.0579|
> |Finite Difference|0.0484|0.3956|0.3137|0.0462|0.1341|0.0663|0.0864|0.0082|-0.0246|0.5402|-0.0589|
> |Absolute Difference|-1.722e-04|-2.090e-03|5.196e-04|-3.385e-04|-1.415e-03|7.247e-04|2.143e-04|-9.613e-05|2.195e-03|2.352e-04|9.514e-04|
> |Element-wise Error|0.0036|0.0053|0.0017|0.0074|0.0107|0.0108|0.0025|0.0119|0.0981|0.0004|0.0164|
>
> $\lVert \text{diff} \rVert$: 0.0036
> Relative Error: 0.0047
>
> **4. Did you consider submitting to the Datasets & Benchmarks track instead of the general conference? Your contribution feels more like a benchmarking suite than machine learning research.**
>
> While our submission does include a suite of tasks for inverse problems in fluid environments, we believe it goes beyond merely being a benchmarking suite. Our paper introduces a novel framework that provides a comprehensive parametric geometry pipeline for specifying complex geometries and offers gradients through the geometry layer. This combination, along with our efficient solver and gradient computation implementation, enables optimization and manipulation tasks on complex shape boundaries that were previously unachievable. These contributions position our work as an advancement in the differentiable simulation community, particularly in its application to robotics, computational fluid dynamics (CFD), and inverse design. Therefore, we believe the general conference track is more appropriate for our submission as it highlights the innovative aspects of our research.
>
>
> **5. In the caption of Table 1, you mention that "# Frames" is not the same as the number of solver steps. What is the actual number of solver steps between two frames? Accumulating gradients over hundreds of steps often leads to exploding gradients, so why don't you have these issues if you stick to the CFL number?**
> * Thank you for your insightful question. The issue of gradient explosion or vanishing is indeed a common problem in differentiable physics. However, we have not observed significant problems in our case, likely due to the clean gradients we provide and the stability of our numerical solver. We provide the statistics of the gradient norm over the full course of optimization on three tasks with varying complexities in Table 1. Our results do not show evidence of gradient explosion or vanishing. In practical engineering scenarios, if a gradient explosion is encountered, we implement gradient clipping and clip the gradient of the network to 1.0. We will include this detail in the implementation section of our paper to enhance the clarity and robustness of our approach.
> * To address your question regarding the number of solver steps, we have provided additional statistics on the actual number of solver steps in the upper part of Table 1 over one full course of optimization on three tasks.
>
> **6. ...using learned surrogates as PDE solvers has demonstrated great potential in overcoming the exploding gradients issue by ... 200x larger time integration steps [1]. Does your framework allow for replacing the NSE solver with a learned surrogate, e.g. a U-Net or FNO?**
>
> Our geometry and optimization framework is designed to be orthogonal to our solver module, allowing for integration with learned NSE solvers. One advantage of our PDE solver over learned surrogates is its ability to solve any given combination of geometry boundary and initial conditions without prior training. In contrast, a learned alternative typically requires extensive training on a large dataset with various boundary conditions and geometries to achieve sufficient generalizability for geometry optimization. The design and control tasks that we highlight in this paper aim to explore novel geometry designs and control in unseen simulation conditions, which a learned surrogate might struggle to do and remains a challenging and active area of research.
> We acknowledge that learned surrogate models have the potential to enhance traditional differentiable physics simulators by enabling larger time steps, and we view this research direction as complementary to our work and a promising area for future development.

---

> > ### Comment · Reviewer_ZAVx · 2024-08-13
> >
> > Thanks a lot for the rebuttal and sorry for the delayed reply. Reviewing 6 papers for NeurIPS has been a ride, and yours is the last one to reply to.
> >
> > Overall reply: I still find this paper an important step toward doing ML on real-world-sized problems, which has been basically impossible with toy tools like PhiFlow. That's why I would keep my score, even if I'm not too happy with the current solver validation.
> >
> > Detailed reply:
> >
> > **1. Inset figures**
> > Thanks.
> >
> > **B. Solver validation**
> > Cylinder flow is a good choice for a validation case. However, looking at the one page rebuttal PDF, the authors obviously don't know much about fluid mechanics. Let me explain.
> > A) "Karman vortex street" is a regime of the cylinder flow, which emerges for Reynolds numbers ($Re=U D/\nu$ with free stream velocity $U$, cylinder diameter $D$ and viscosity $\nu$) between roughly 40 and 1000. Thus, providing $\nu$ in the plots is pretty much useless unless you provide $U$ and $D$ in addition or just provide $Re$.
> > B) Validating a solver does not mean just running it and being impressed by the beauty of the images. One has to 1. configure a case for which we have a reference solution, 2. extract some statistics from the simulation for which we have this reference data, and 3. make sure that the new solver can recover this reference solution. You propose validating in the Karman vortex street regime, so you would need to set up a simulation with say $Re=50$, for which you can find a reference Strouhal number of 0.13 in [Irvine 1999](https://www.semanticscholar.org/paper/KARMAN-VORTEX-SHEDDING-AND-THE-STROUHAL-NUMBER-Irvine/181f6da07be036ac3de89e21f64ebedd28f68a90). But of course you would have to first extract this Strouhal number (which is related to the frequency of the shading) from your simulation. I hope now you understand why people in numerics spend whole PhDs on these things, and these are must-haves for credibility.
> > C) Now, back to your rebuttal Figure 1: viscosity $\nu=0$ means $Re=\inf$, which means chaotic turbulence, which is not what your figure shows.
> >
> > **3. Gradient validation**
> > Looks good!
> >
> > **4. Benchmark track?**
> > Ok, I see the point.
> >
> > **5. # Frames to differentiate through**
> > This part sounds almost too good to be true, but I'm not an expert in gradient stability. I hope there is nothing wrong with your code, but adding some more details on why you don't have these gradient issues might be helpful.
> >
> > **6. Flexibility of framework to use other PDE solvers**
> > Ok, sounds good.

---

> ### Author Response · Authors · 2024-08-12
>
> Dear reviewer ZAVx, we wanted to check if there are any remaining questions or concerns about our rebuttal. Since the author-reviewer discussion period ends tomorrow, we’d appreciate any final feedback you might have.
>
> Thank you for your time and attention.

---

> ### Author Response · Authors · 2024-08-13
>
> Thank you for your detailed feedback and for providing a thorough suggestion of the validation process in fluid mechanics.
>
> Regarding your points:
>
> 1. **Numerical Viscosity in Semi-Lagrangian Scheme**: We acknowledge that our semi-Lagrangian scheme introduces numerical dissipation [1], which explains the results in Figure 1 in the rebuttal PDF for the chaotic turbulence case. We will add this in the discussion section of our method.
>
> 2. **Validation with Reference Data**: Thank you for the additional feedback. To further validate our solver numerically, we focused on validating the mid-viscosity scenario that we provided in the rebuttal PDF, corresponding to a Reynolds number of 209. For this case, we calculated a Strouhal number of 0.16, compared to the reference value of 0.18 from Irvine 1999. The observed discrepancy can be attributed to the numerical dissipation introduced by our scheme.
>
> We appreciate your insights, which have helped us better articulate the limitations and implications of our approach.  Please let us know if you have further questions.
>
> [1] Jos Stam. 1999. Stable fluids. In Proceedings of the 26th annual conference on Computer graphics and interactive techniques (SIGGRAPH '99). ACM Press/Addison-Wesley Publishing Co., USA, 121–128. https://doi.org/10.1145/311535.311548

---

### Official Review · Reviewer_grLC · 2024-07-15

**Soundness:** 3
**Presentation:** 3
**Contribution:** 3
**Rating:** 7
**Confidence:** 4

**Summary:**

The paper aims at a fully automated pipeline for devising neural controls for complex fluidic systems with dynamic boundaries. The system consist of externally driven soft boundaries and internal complex flow behaviors.

The proposed framework contains a differentiable geometry representation, a differentiable fluid simulator with solid-fluid interface handling, and a control-shape co-design algorithm using gradient-based optimization.
-  The 3D surface is represented by a list of 2D surfaces (represented by a set of connected Bezier curves of their control points) defining the first and last cross-section, and each key cross-sections of the geometry. More complex 3D geometries are represented by union and intersection of sub-geometries.
- The fluid simulator leverages the operator-splitting method. A single simulation step comprises three sub-steps: advection, viscosity, and projection. It uses MAC grid.
- The back-propagation process is used to construct the gradients of the geometry surface w.r.t. the loss function.

The method is evaluated on multiple tasks, from simpler 2D tasks to complex 3D tasks (such as,
neural gate controller and artificial heart design).

**Strengths:**

- The paper contains a fair amount of work to develop a differentiable fluid simulation and optimization framework, which includes a differentiable geometry representation, a differentiable fluid simulator with solid-fluid interface handling, and a gradient-based optimization framework.
- The proposed methods are evaluated on multiple cases from simple 2D examples to complex 3D tasks. Besides, it's nice to see the performance profiling and comparison results in the paper.

**Weaknesses:**

It would be nice to see the framework open-sourced.

**Questions:**

- It might be useful to demonstrate whether gradients explode or vanish during the iterative back-propagation process.
- In the experiment studies the effect of random initialization in our fluid optimization tasks, the Shape Identifier task (with #Parameter = 10) is used. Does the same conclusion hold for more complex tasks with a greater number of parameters?
- How sensitive are the design results to the grid resolution? Given that the grid used for generating the design is relatively small for fluid simulations, can the designed system achieve the same level of accuracy when evaluated on a higher resolution grid?
- How does the method perform when using a higher resolution grid (but not so high that the simulator cannot finish within an acceptable time)?
- In Table 2: Time Performance, what's the time unit?

**Limitations:**

The author have discussed the limitations in the paper.

---

> ### Author Rebuttal · Authors · 2024-08-07
>
> We thank the reviewer for the constructive suggestions. Below we provide responses to individual questions.
>
> **1. It would be nice to see the framework open-sourced.**
>
> We will open-source our code upon acceptance. We have provided the anonymized version of our code at https://anonymous.4open.science/r/DiffFluidRebuttal-1D24/.
>
> **2. It might be useful to demonstrate whether gradients explode or vanish during the iterative back-propagation process.**
>
>  * Thank you for your insightful point. The issue of gradient explosion or vanishing is indeed a common problem in differentiable physics. However, we have not observed significant problems in our case, likely due to the clean gradients we provide and the stability of our numerical solver.
>  * We have included additional statistics (min,max,mean, standard deviation) on the gradient norm over the full course of optimization for three tasks with varying complexities in Table 1. Our results do not show evidence of gradient explosion or vanishing.
> * In practice, in case a gradient explosion is encountered, we implement gradient clipping of the gradient of the network to 1.0. We will include this detail in the implementation section of our paper to enhance the clarity and robustness of our approach.
>
>
> **3. In the experiment studies the effect of random initialization in your fluid optimization tasks, the Shape Identifier task (with #Parameter = 10) is used. Does the same conclusion hold for more complex tasks with a greater number of parameters?**
>
> Yes, the same conclusion holds for all of our tasks. In Fig 2 we present an additional verification with the 3D heart controller task which has 7.1k neural network parameters. We initialize the network parameters with 5 different random seeds, and observe the same convergence across seeds even with different initial behaviors, showing the robustness of our method.
>
> **4. How sensitive are the design results to the grid resolution? Given that the grid used for generating the design is relatively small for fluid simulations, can the designed system achieve the same level of accuracy when evaluated on a higher resolution grid? How does the method perform when using a higher resolution grid (but not so high that the simulator cannot finish within an acceptable time)?**
>
> * Thank you for raising this important question. In pure forward simulation, higher grid resolutions are necessary to achieve greater accuracy, particularly at the solid-fluid interface, which directly impacts optimization results. This is because the same continuous geometry boundary can yield different simulation results depending on the resolution, the objective values—and consequently the optimized designs—will vary across different resolutions.
> * To illustrate this, we have conducted an additional experiment to analyze the effect of grid resolution on design outcomes, where we first optimize for the Shape Identifier task under 128x128, then take the optimized values to 512x512, which evaluates to a loss value of 3.11. In comparison, a randomly initialized design under 512x512 evaluates to a loss value of 245.1 and optimizes to a design with loss 0.968.  The results demonstrate that while higher resolutions generally improve accuracy, they also introduce variations in the optimal design due to the increased detail captured at the boundary.
>
>
> **5. In Table 2: Time Performance, what's the time unit?**
>
> The time reported is in Seconds (s)

---

> ### Author Response · Authors · 2024-08-12
>
> Dear reviewer grLC, we wanted to check if there are any remaining questions or concerns about our rebuttal. Since the author-reviewer discussion period ends tomorrow, we’d appreciate any final feedback you might have.
>
> Thank you for your time and attention.

---

> ### Author Response · Authors · 2024-08-13
>
> Dear Reviewer,
>
> As the discussion deadline approaches, we are eager to address any remaining concerns you may have. We kindly request you to review our responses. If there is anything further we can do to facilitate the evaluation process, please let us know. We also respectfully ask that you reconsider our score based on the rebuttal provided.
>
> Thank you for your attention.
>
> Best regards,
>
> Authors

---

> ### Comment · Reviewer_grLC · 2024-08-14
>
> I thank the authors for the detailed response. Overall, I think this is a solid paper and I would like to keep my rating of "accept".

---

### Author Rebuttal · Authors · 2024-08-07

We thank all reviewers and the AC for their time and effort in reviewing and for insightful comments to strengthen our work. Besides the responses to individual reviewers, here we would like to highlight our contributions and new quantitative/qualitative results added in the rebuttal.

1. **Contributions**
    1. **[Motivation]** Our manuscript tackles an important, challenging, and significant problem in the optimal design and control of fluid flows in complex contexts [Reviewer ueL6] and effectively bridges the gap between toy experiments and real-world problems [Reviewer ZAVx].
    2. **[Method]** Our method is novel, implementing an efficient PDE solver from scratch in C++ and CUDA with analytically derived gradients, providing a great service to the community [Reviewer ZAVx, Reviewer grLC]. The differentiable fluid simulation and optimization framework includes a differentiable geometry representation and fluid simulator with solid-fluid interface handling [Reviewer grLC].
    3. **[Experiments]** Our extensive experiments cover a range of scenarios from simple 2D examples to complex,challenging 3D tasks [Reviewer grLC], not just usual "toy" problems [Reviewer ueL6]. We also provide performance profiling and comparison results [Reviewer grLC] and highlights the advantages gradient-based optimizations compared to gradient-free methods [Reviewer QmuX].
    4. **[Presentation]** Our manuscript is well-written, and provides a good balance between theory and experimental evidence [Reviewer ZAVx, Reviewer grLC].


2. **New Results**
    1. **[Code: Open-Source Release]**
    https://anonymous.4open.science/r/DiffFluidRebuttal-1D24/ We have anonymously released our code open-source for the review stage and will officially release it upon acceptance, enabling the community to reproduce our results and build upon our work.
    2. **[Experiment: Solver Validation]**. We validated our fluid solver using the classic Karman Vortex Street test.
    3. **[Experiment: Gradient Validation]**. We presented a gradient validation test to compare our analytical gradient to finite-difference gradient.
    4. **[Experiment: Gradient Norm Analysis]**. We provide statistics on gradient norm on multiple tasks during optimization to study whether gradients diminish/explode.
    5. **[Experiment: Step Number Statistics]**. We provide statistics for solver steps on multiple tasks.
    6. **[Experiment: Effect of Initialization on Complex Tasks]**. We provide an additional experiment on the effect of initialization on the 3D neural controller task with 7k parameters.
    7. **[Experiment: Memory & Time Comparison with DiffTaichi]**. We compare the memory usage and computational time of our solver with DiffTaichi, highlighting the advantages of our implementation in terms of efficiency and scalability.
    8. **[Experiment: Effect of Grid Resolution on Optimization]**. We explore how varying grid resolutions impact the optimization outcomes.
    9. **[Implementation Details: GPU Code Framework & Kernel Acceleration]**. We detailed our GPU code framework and kernel acceleration techniques.

---

### Author Response · Authors · 2024-08-07
**Clarifications of the Attached Codebase**

Dear AC,

In response to the requests from the reviewers, we have shared our codebase through the anonymous link - https://anonymous.4open.science/r/DiffFluidRebuttal-1D24/. Please kindly confirm if this is the preferred way to share it.

We appreciate the reviewers' interest and attention to our manuscript and detailed implementation. Thank you for your valuable feedback.

---

> ### Comment · Area_Chair_SnuM · 2024-08-10
>
> Dear Authors,
>
> Yes, that is appropriate.
>
> Best,\
> AC

---

### Decision · Program_Chairs · 2024-09-25

**Decision:**

Accept (poster)

**Comment:**

The paper describes a neural network-based framework for the design and control of fluidic systems that involve dynamic (deformable) boundaries. The framework includes a differentiable Navier-Stokes solver, a parametric (Bezier) representation of the geometry, and an algorithm for joint optimization of control and design. The optimization and evaluation utilize a proposed gym-like RL environment along with a series benchmark experimental scenarios. Experiments, including those that involve the proposed benchmarks, reveal the advantages of the framework.

The paper was reviewed by four referees whose overall assessments of the paper all lean positive. Several reviewers find that the problem is interesting and that the algorithmic implementation (i.e., the C++/CUDA PDE solver and the Bezier parameterization) provide a valuable contribution to the community. Others emphasize the thoroughness of the experimental evaluation. There were some concerns about the significance of backpropagation in time, with one reviewer commenting that it has been explored extensively for neural ODE solvers. In this way, the two reviewers find that the contributions relative to existing work is not sufficiently clear (i.e., with regard to the CUDA implementation). The same reviewer emphasizes that the paper does not adequately discuss the limitations encountered during backpropagation with regard to simulating differentiable fluids. In their response to the reviewers, the authors provide more detail regarding their CUDA kernel implementation, which helps to clarify the contribution as the implementation of a custom fluidic solver, as well as elaborate on issues related to backpropagation. However, as the reviewer points out, the author response could have better conveyed the paper's contributions and its relevance to the community. Meanwhile, other reviewers comment on the need to provide a validation of the solver, which the authors do in their rebuttal, and to make the code publicly available, which the authors clarify will be done if and when the paper is published.

Overall, the quality of the paper improved as a result of the initial reviews and the author feedback, as indicated by the more favorable reviewers maintaining their high scores and the two less favorable reviewers increasing their score. The AC agrees with Reviewer QmuX that the paper could be improved further by being more explicit about the contributions relative to existing methods and the relevance to the community.